# Insights from a Case of Good’s Syndrome (Immunodeficiency with Thymoma)

**DOI:** 10.3390/biomedicines11061605

**Published:** 2023-06-01

**Authors:** Roberto Paganelli, Michela Di Lizia, Marika D’Urbano, Alessia Gatta, Alessia Paganelli, Paolo Amerio, Paola Parronchi

**Affiliations:** 1Department of Medicine and Sciences of Aging, University “G. D’Annunzio” of Chieti-Pescara, 66100 Chieti, Italy; p.amerio@unich.it; 2Internal Medicine, School of Medicine, UniCamillus, Saint Camillus International University of Health Sciences, 00131 Rome, Italy; 3Allergology ASL Teramo, Hospital of Giulianova, 64021 Giulianova, Italy; michela.dilizia@aslteramo.it; 4Laboratory Unit, Hospital S. Annunziata, 67039 Sulmona, Italy; delta.mea@virgilio.it; 5Allergology Service, ASL Chieti, 66100 Chieti, Italy; alessiagatta@ymail.it; 6PhD Course in Clinical and Experimental Medicine, University of Modena-Reggio Emilia, 41121 Modena, Italy; alessia.paganelli@gmail.com; 7Department of Experimental Medicine, University of Florence, 50121 Florence, Italy; paola.parronchi@unifi.it

**Keywords:** Good’s syndrome, thymoma, hypogammaglobulinemia, cellular immunity, immunophenotype, autoantibodies

## Abstract

Immunodeficiency with thymoma was described by R.A. Good in 1954 and is also named after him. The syndrome is characterized by hypogammaglobulinemia associated with thymoma and recurrent infections, bacterial but also viral, fungal and parasitic. Autoimmune diseases, mainly pure red cell aplasia, other hematological disorders and erosive lichen planus are a common finding. We describe here a typical case exhibiting all these clinical features and report a detailed immunophenotypic assessment, as well as the positivity for autoantibodies against three cytokines (IFN-alpha, IL-6 and GM-CSF), which may add to known immune abnormalities. A review of the published literature, based on case series and immunological studies, offers some hints on the still unsolved issues of this rare condition.

## 1. Introduction

In 1954, the late Robert A. Good (b.1922–d.2003) in collaboration with Dr. R.L. Varco, described a type of agammaglobulinemia associated with the presence of a tumor of the thymus [1,2], the mediastinal gland that Dr. Good greatly contributed to recognize as the central organ for the development of the immune system [3,4,5]. It was only in the early ‘60s that the thymic derived lymphocytes were shown to be responsible for delayed type reactions and transplant rejection [6], and later that two distinct interacting subtypes of lymphocytes—T and B cells—were present also in mammals, the latter being responsible for antibody formation [6,7].

The word thymoma has been used to indicate a variety of neoplasms originating in the thymus; however, many tumors arising in the thymus differ both clinically and pathologically. Their distinction was established only in the late ‘70s, distinguishing thymomas from both lymphomas and carcinomas [8,9]. Thymomas are regarded as epithelial tumors, described by the WHO classification in 1999 [10]. It was well known that myasthenia gravis was associated with a thymoma since 1950 [11], and at the time of R.A. Good’s description, nothing could link this thymic neoplasia to agammaglobulinemia. This association was rapidly known as Good’s syndrome [12], and several case and series reports have been published until today: we retrieved 154 papers with “Good’s syndrome” in their title in Pub Med. Despite being diagnosed mainly in adults (with median age at diagnosis 55–64 years in a recent review [13]), it was included in the primary immune deficiencies, with the name of Immunodeficiency with thymoma (sometimes changed to thymoma with immunodeficiency) [14,15,16], which is its official designation. In the 2019 the International Union of Immunological Societies (IUIS) classification of Inborn Errors of Immunity [16] moved it in group X, containing Phenocopies of PIDs (primary immunodeficiencies). We shall come back to this point in the discussion.

The association of thymoma and hypogammaglobulinemia was recognized as having a worse prognosis than other types of hypogammaglobulinemia, since not only bacterial but also viral, fungal and parasitic infections were reported [17,18,19,20,21,22]. These were opportunistic infections which very rarely occurred in agammaglobulinemia, pointing to the presence of defects also in the cellular compartment of the immune response, dependent on thymus-derived lymphocytes. Moreover, other characteristic disorders accompanied the syndrome, most commonly pure red cell aplasia [23,24] and lichen planus [25,26,27]. Another typical aspect of this rare syndrome is the presence of chronic diarrhea [28,29], myelodysplastic syndrome [24], pancytopenia [30] and an association with autoimmune manifestations such as myasthenia gravis and pure red cell aplasia [13,31]. Thymomas are known to be associated with autoimmune disorders also in the absence of hypo-agammaglobulinemia [32,33,34,35], such as polyglandular syndrome, enteropathy and hemolytic anemia, sometimes described as paraneoplastic manifestations [36]. Thymectomy does not represent a useful treatment for these autoimmune disorders; however, it is superior to conservative medical therapy in myasthenia gravis uncomplicated by comorbidities [37,38,39,40], and it is recommended in most cases of myasthenia, in particular the ocular form [37] and juvenile-onset [41]. Recently more attention has been devoted to the type of surgery and the factors affecting the outcome [39,42]; however, complete remission is achieved by only one-third of cases [43]. Among the adverse effects of thymectomy, the appearance of organ and non-organ specific autoantibodies [44] and increased risk for systemic autoimmune diseases [45] have been highlighted. Moreover, thymus derived B cells and plasmablasts persist in tissues after thymectomy [46]. In the case of Good’s syndrome, thymectomy does not provide a better prognosis; neither does it restore immunological function [47,48]. These considerations argue against the paraneoplastic nature of Good’s syndrome and all its associated manifestations. 

Many different hypotheses on the pathogenesis of Good’s syndrome have been advanced, but so far none has gained acceptance; therefore, this syndrome “still remains a mystery” [49] approaching 70 years after its original description.

Here, we report the case of a patient we have followed for the past ten years and who died in 2022. Some immunologic findings obtained may be of relevance for the interpretation of the pathogenesis and the clinical course of this syndrome.

## 2. Case Description

Male subject, of Caucasian ethnicity, born in 1968. He recalls suffering from chronic sinusitis in young age and was diagnosed with hiatal hernia and erosive atrophic gastritis in 2003 (gastroscopy revealed also H. pylori infection, successfully eradicated). In 2004, the patient was admitted to the hematology ward of the Hospital in Pescara where he was diagnosed with warm antibody hemolytic anemia, Coombs positive. He was discharged with successful prednisone treatment for 6 months. A bone marrow biopsy revealed no abnormality. In mid-2008 he suffered from protracted diarrhea (3–6 times daily, without blood) without fever. A stool exam tested positive for *Giardia lamblia*, persistent after repeated cycles of antibiotics. A breath test showed lactose intolerance, but no benefit was obtained with a dairy-free diet for six months. Stool cultures became negative for parasites, but occasionally positive for *Clostridium difficile*, *Salmonella* spp. And *Shigella* from 2009 to 2014. After two years of persistent diarrhea, without benefit from attempted treatments, the patient is referred to an allergist on the suspicion of a food intolerance. In 2010, he is tested for the first time for serum immunoglobulins, finding severe hypogammaglobulinemia (IgG 27 mg/dL). After a negative result for amyloidosis on periumbilical fat, a provisional diagnosis of Common variable immunodeficiency is made at the Policlinico of Careggi Hospital in Florence, and in 2011 i.v. Ig replacement therapy is started at 30 g every three weeks (400 mg/kg/month). During the same year, the patient undergoes cholecystectomy for biliary lithiasis and suffers from esophageal candidiasis, responding to fluconazole treatment. In 2012, biopsies taken during colonoscopy reveal the presence of cytomegalovirus (CMV), and high levels of circulating CMV DNA (2 × 10^5^ copies/mL) are found. Admitted to the hematology ward, the patient is treated with Gancyclovir, which reduced CMV DNA (still 277 copies/mL detected in 2014), but diarrheal symptoms persisted. An eye specialist visit ruled out the presence of CMV retinitis, and human immunodeficiency virus (HIV) tests scored repeatedly negative.

An abdominal ultrasound in 2009 demonstrated mild splenomegaly (14 × 6.5 cm), gallbladder lithiasis and enlarged liver, with multiple mesenteric lymphadenopathies with diameters sometimes larger than 1 cm. Non-specific ileitis was observed at colonoscopy, with villous pseudo-atrophy without intraepithelial lymphocytic infiltrates. The lamina propria had marked gland hyperplasia with increased inflammatory infiltrate (lympho-plasmacellular, with neutrophils and eosinophils, but no crypt abscesses, possibly due to Crohn’s disease or eosinophilic enteritis). Fecal calprotectin was negative, as well as serology for celiac disease. In 2008, the patient underwent surgical removal of a lesion on the tongue and oral mucosa, diagnosed histologically as an area of inflamed leukoplakia.

A control chest X-ray revealed the presence of a pseudocapsulated multilobular mediastinal mass dimensions 6.1 × 3.4 cm, possibly a thymoma, and in 2012 it was surgically removed in thoracoscopy, with thymus preservation. The histological examination showed epithelial-like infiltrates with large elements aggregates, some with spindle shaped aspect, thin chromatin with large nucleoli and cytoplasm; final diagnosis was a medullary thymoma, type A according to the World Health Organization (WHO) classification (1999), with capsular infiltration but not present in the surgical resection margins.

In 2013, the patient is admitted to the infectious disease hospital for persistent diarrhea and diagnosed with proctocolitis with chronic diarrhea and malabsorption, prescribed treatment with mesalazine and oral budesonide, leading to partial transient improvement of the symptoms. During the same hospitalization, for recurrence of oral candidiasis, fluconazole and then itraconazole were administered, with diarrhea worsening; amphotericin B was substituted, with little improvement. In 2014 at follow-up, a chest X-ray found a hypodense lesion with demarcated margins, which a PET-CT localized as increased metabolic activity in the medial section of the upper right pulmonary lobe. Therefore, in July 2014 the patient underwent a partial resection of the upper right lobe with removal of the mediastinal mass and thymic residues. The histopathological examination could not find neoplastic cells in both thymus and lung parenchyma, which was filled by hemorrhage relics with bronchiectasis and an intraparenchymal lymph node with normal architecture.

In September 2014, a leukokeratotic lesion developed in the gingival part of the oral cavity and on the tongue was removed. The histology demonstrates a subepithelial infiltrate of neutrophils and lymphocytes, with dermo-epidermal detachment, and a diagnosis of erosive lichen planus is made. At this time, the patient is referred to the immunology center in Chieti, closer to the patient’s city, where s.c. Ig therapy with Hyzentra is started at 8 g weekly. A complete assessment of the immunological status was also made (Table 1). Serum levels of immunoglobulins at the beginning of 2016 were IgG 425 mg/dL, IgA 25 mg/dl and IgM 17 mg/dl. Local treatment with clobetasol three times/day was recommended for the oral pathology, and a gluten-free diet with cycles of oral budesonide for diarrheal symptoms was additionally recommended, which however persisted with only minimal benefit. Lymphocyte counts dropped below 1 × 10^9^/L in 2018, and severe anemia developed. The patient was referred to the hematology clinic in Pescara where despite all treatments a progression of symptoms was observed. Bone marrow failure and sepsis were cause of death in 2022, at age 54.

The most frequent clinical features of patients with Good’s syndrome reported in the literature (Table 2) are listed, and the main clinical findings of our case, in order of time of manifestation, are shown in Table 3.

## 3. Immunological Evaluation

### 3.1. Lymphocyte Subpopulations

The patient was evaluated on three different occasions during his attendance of the Clinic in Chieti. The overall results did not vary significantly despite being on Ig replacement therapy. Data are reported in Table 1 as absolute numbers derived from full blood counts, in order to have uniform data. The cytofluorimetric analysis was carried out on EDTA-anticoagulated blood samples stained in double, triple or quadruple fluorescence with the following monoclonal antibodies (all from Beckton-Dickinson, Mt. View, CA, USA): CD3, CD4, CD8, CD45RA, CD27, CD25, CD28, CD31, CD19, CD20, CD21, CD24, CD38, CD16, CD56 and human IgD, conjugated with different fluorochromes.

Results are reported in Table 1. The main abnormalities found were lymphopenia (absolute numbers/μL 932–1138 lymphocytes; normal values > 1500), with very low CD19+ B cells (2–15 cells/μL; normal values 65–700) and very low CD4+ T cell numbers (289–296/μL; normal values 500–1800). There was an inverted CD4/CD8 ratio of 0.45 (normal > 1.1), with CD8+ T cells 636/μL which is a slightly elevated value compared to normal (250–660/μL). Other findings included the absence of CD21+ B cells, low expression of the CD19 and CD20 markers (limited to 0.6 and 1.4% of lymphocytes, respectively) and near absent expression of IgD on the surface (0.14% IgD+), which indicates a B cell population depleted of unswitched cells. In addition, there is absence of B cells coexpressing CD38 and CD24 (transitional B lymphocytes) but a presence of CD38 + CD24- plasmablasts (usually very rare in unstimulated condition) and low CD16 + CD56 + CD3- (putative NK) cells (52/μL; normal values 27–450). Among CD4+ T cells, we found low naïve phenotype CD4+ (30% CD45RA+, usual for the patient’s age) with near absent recent thymic emigrants (0.4% CD4 + CD31+); we also noted that CD45RA+ cells were two-thirds of CD8+ T lymphocytes, in contrast to what found for CD4+ T cells, and increased proportion of CD8+ CD28- (68.6%), which are functionally exhausted cytotoxic T lymphocytes (see results with absolute numbers/μL in Table 1). We also noted an increase of double negative CD3+ T cells and the lack of memory CD27+ B lymphocytes which is at variance with the near absence of unswitched ones, indicated by low CD27+ and IgD+ cells. Some of these abnormalities in lymphocytes subpopulations have not been previously described in patients with Good’s syndrome.

### 3.2. Autoantibody Detection

Some 96-well microtiter vinyl plates were coated overnight with 10 ug/mL of each recombinant cytokine tested, diluted in carbonate buffer pH 9.6. After washing with 0.15 M NaCl solution containing 0.05% Tween 20 and 0.1% *w*/*v* bovine serum albumin, patient serum, along with control serum, diluted 1:10 was dispensed to triplicate wells and left to incubate for 3 h at room temperature. After three more washings, the bound antibody was revealed by anti-human IgG antiserum, conjugated with horse radish peroxidase, and developed after 1 h with substrate in the dark. The color reaction was read at 492 nm. All products were purchased through Sigma Chem Co (St Louis, MO, USA). Positive results were those with optical density three times the average mean of normal human serum. Two samples from the patient were tested, one collected in 2014 before lobe resection with removal of thymic residues and the other in 2017 when severe anemia developed, and they were stored at −70 °C in aliquots, thawed only once. The results were qualitatively identical, with no changes for positive and negative tests.

Autoantibodies directed against interferon (IFN)-α (high), GM-CSF and IL-6 (low positivity) were detected in both samples (above three standard deviations from healthy donors’ values), whereas they tested negative for antibodies to IFN-γ, IL-8 and TNF-α. Not all these specific autoantibodies have been studied previously. We could not assess antibodies to erythropoietic factors, IL-7, IL-17 and other relevant cytokines.

## 4. Discussion

Over time, Good’s syndrome has been considered a variant of common variable immunodeficiency (acquired agammaglobulinemia), soon differentiated from other types because of its nature of combined defect of both B and T cells [50], a paraneoplastic syndrome and finally a thymoma-associated autoimmune disorder, which presents autoantibodies to cytokines and growth factors, therefore mimicking other primary immunodeficiencies (phenocopy). The very nature of this disorder, whether genetically determined (primary) or secondary to cancer or more probably to infection [51], has long been debated, without a clear conclusion. However, none of these classifications, which presume to identify the underlying defect able to account for all the clinical aspects of the syndrome, seem to be completely satisfactory.

### 4.1. Review of Literature

The heterogeneity of both the clinical presentation and the immunological abnormalities of Good’s syndrome has been repeatedly underlined (see Table 4). In early reports [12] with only 30 cases published, the picture was dominated by sinopulmonary infections and hypogammaglobulinemia, with mention of the rarity of pure red cell aplasia or pancytopenia, and other features such as chronic diarrhea with malabsorption and also fungal infections. The underlying mechanism of these symptoms was unknown, since at the time the precise role of the thymus in the adult immune system had not been adequately defined [12], but the presence of defects of delayed hypersensitivity was noted. Only a few reports of large case series have been published. In 2001, Tarr and colleagues [52], describing five new cases, searched the literature and reported on a total 51 patients with Good’s syndrome. Their cases showed, in addition to recurrent chest and sinus bacterial infections, also *Pneumocystis* pneumonia, frequent CMV visceral infection, both infectious (*C. jejunii, E. coli*) and non-infectious colitis with diarrhea and, among the immunologic findings, the absence or low numbers of B cells, CD4 lymphopenia with inverted CD4/CD8 ratios and in one case reduced NK cells [52]. The overall clinical picture of the 51 cases, aged 56 on average when first symptoms developed and 62 years when thymoma and hypogammaglobulinemia were diagnosed, was dominated by Hemophilus influenzae infection along with other bacterial pathogens. Eleven patients had mucocutaneous or esophageal candidiasis; five had CMV infection, with seven more where CMV was detected without a clear significance. Chronic diarrhea, mainly of unknown cause, was present in 19 patients, and no mention of lichen planus or autoimmune manifestations was made. Even the immunologic findings are equivocal at best, with B cell lymphopenia reported in 9/11 tested and 50% of patients having normal CD4 and CD8 T cells (12/25 measured). In this review, 29/51 patients had died during the follow-up, another clue that the syndrome had a worse prognosis than common variable immunodeficiency. Jansen et al. in 2016 [53] reported a mortality of 16/39 cases identified both in the clinical centers and in the European Society for ImmunoDeficiencies (ESID) registry with a mean 9 years’ follow-up and a median survival of 14 years from diagnosis. The hazard ratio compared to the life table for European countries was significant (3.7 *p* = 0.008). Treatments were >90% immunoglobulin supplementation and antibiotic prophylaxis in 63%, with 8.5% receiving G-CSF for hematologic involvement. Twenty-four (51%) cases had concomitant autoimmune disease, mostly pure red cell aplasia and lichen planus.

Kelleher and Misbah (2003) [2] reported that about 1% of immunoglobulin deficiencies might be diagnosed as Good’s syndrome and that hypogammaglobulinemia was associated with 6–11% of thymomas. In their series of 75 cases from those published, they highlight the frequency of hematologic disorders (50% red cell aplasia, 55% leukopenia) and viral infections (about 40% of cases, mainly CMV), despite the heterogeneity of immunological findings (reduced B cells in 33/38, inverted CD4/CD8 ratio in 22/24, low CD4+ T cells in 9/20).

In 2007, Agarwal and Cunningham-Rundles [54] described two unusual cases, one with babesiosis and the other with Kaposi’s sarcoma, and advocated the use of the term “immunodeficiency with thymoma” because the patients had several other immunological defects in addition to hypogammaglobulinemia. Among the possible pathogenetic mechanisms, they mentioned the dysregulated production of limitin (now known as IFN-zeta [55]) a potent stromal-derived inhibitor of B-cell lymphopoiesis [56], which has been also included in a recent review [51]. It was reported in 1981 that T cells from patients with thymoma and immunodeficiency suppress in vitro normal pre-B as well as B lymphocyte differentiation [57], and both erythropoiesis and immunoglobulin production were described to be suppressed by T lymphocytes [58]. Two genetic mutations have been identified involving members of the Tumor Necrosis Factor Receptor superfamily, transmembrane activator and CAML interactor (TACI) and BAFF-R [59,60,61] factors known to promote B cell survival and differentiation [62]. These mutations, which are associated with common variable immunodeficiency in about 10% of cases [63], have been found also in unaffected individuals and are currently thought to represent predisposing factors to B cell defects.

Kelesidis et al. in 2010 [49] published a review comprising 152 patients with Good’s syndrome and included the following defining aspects of immunodeficiency: hypogammaglobulinemia, low or absent B cells, variable defects in cell-mediated immunity with a CD4 T lymphopenia, an inverted CD4/CD8+ T-cell ratio and reduced T-cell mitogen proliferative responses. Additionally, in their review, mortality during the report follow-up was high (44%) with an overall mortality of 46%, the type of non-bacterial infectious agents included CMV and Candida but also Herpes zoster, Herpes simplex, Pneumocystis and Giardia lamblia in four or more instances. Diarrhea was present in 31.8%, mostly non-infectious. Absent delayed hypersensitivity and mitogen responsiveness were found in 60% of cases. The presence of autoantibodies in many cases (including antinuclear, anti-striated muscle and anti-thymic) suggested an autoimmune pathogenesis, despite lack of correction following either corticosteroid treatment or thymectomy. The presence of autoimmune regulator (AIRE) gene defects has not been studied in this syndrome, despite its expression in the thymus and its role in negative selection [64]; moreover, anti-interferon type I autoantibodies have been found in Autoimmune-poly-endocrinopathy-candidiasis-ectodermal dystrophy (APECED) [65], caused by AIRE mutations [66]. Other autoantibodies to cytokines have been found in this syndrome and also in mucocutaneous candidiasis [67,68].

In 2017, the only non-Western systematic review was published by Dong et al. [69], including 47 patients, 31 females, with mean age of 55, diagnosed in China. The main differences of this series with the previous ones are the predominant sinopulmonary infections with Pseudomonas and Klebsiella, but not Hemophilus, and the low B and CD4+ cells in 100% and 95% of cases, respectively. Of these Chinese cases, 36% had diarrhea, and 36% had autoimmune manifestations. Notably, four patients suffered hearing loss, and leukopenia was detected in 55%, but leukocytosis was detected in 23%. A low NK cell number was found in 92%, and high CD8+ T cells were found in 50%.

In their 2018 review of 78 cases from the UK-Primary Immune Deficiency, Zaman and coworkers [70] confirmed many of the previously described findings, but with younger age at presentation (median 54 years), a higher proportion of females (59%), lower mortality (only 9%) and autoimmune diseases (26%). The most common type of thymoma (see Table 4) was AB, with only four type A (medullary) cases. Type C (malignant) cases also had severe lymphopenia, but in the other types a mere 15% had CD4+ cell <250/μL. All patients had absence of B lymphocytes and were receiving immunoglobulins for their hypogammaglobulinemia.

Shi and Wang (2021) made a review of 162 patients, and they attempted also to define subgroups of this heterogeneous syndrome, by a clustering analysis [13]. Many findings were similar to previous reports: age at diagnosis 58 years, 10-year survival 53.7%, with significant prognostic factors such as thymoma status, infection associated with cellular immunity defect, sinopulmonary and central nerve system and bloodstream infections. Associated autoimmune features were pure red cell aplasia in 33%, myasthenia gravis in 27% and lichen planus in 23%. This latter manifestation has been reviewed recently [27]. Three clusters were proposed, based on predominance of infections related to either cellular immunity defects (51 cases), humoral immunodeficiency (34 cases) or other (other or unknown pathogens) defects (77 cases). The cellular defect group had a significantly higher mortality.

In a very recent analysis, Kabir et al. [51] underlined that Good’s syndrome is very different from other types of adult hypogammaglobulinemia, and it has features of combined immunodeficiency [13]. B cell progenitors in the bone marrow seem to be arrested at an early stage. However, there is no association between thymoma type (Table 4) and the type of opportunistic infections or autoimmune complications. Since the cause and defects underlying thymoma development have not been elucidated, it is only a speculation that thymic tumor microenvironment can cause aberrant maturation of autoreactive T cells; however, oligoclonal expansions of bone marrow resident beta variable region 8-CD8+T cells have been reported in some cases [71], but no corresponding population was found in the periphery.

A summary of phenotyping of peripheral immune cells has been published by Ternavasio-de La Vega et al. in 2011 [48], showing persistence of alterations after thymectomy. In a short review, they included the following findings: persistent low B cells, with absence of memory B; normalization of CD4/CD8 ratio after thymectomy, but the absence of memory CD8+ T cells (no CD45RA-) and circulating dendritic cells, normal NK (CD56+). The patient studied had normal CD4+ lymphocytes from the start but very low B (0.18%).

Table 4 shows a comparison of thymoma types in large series of patients with Good’s syndrome, according to the WHO classification. Statistical analysis based on contingency tables for values were indicated by asterisk*. Only a fraction of total patients were reported for thymoma type.

### 4.2. Comment on Our Case Findings

The patient we followed had a type A (medullary) thymoma, which was resected in two times (the second one for thymic non-malignant remnants, together with lobectomy), which is associated with a benign prognosis, but this was not the case for associated complications, infectious, hematological and autoimmune or other (oral erosive lichen planus and intractable diarrhea). Bone marrow failure was not aggressively treated with immunosuppressive agents, but immunoglobulin replacement therapy was adequate, and antibiotic prophylaxis provided. The peripheral blood phenotyping was performed extensively and revealed profound imbalances. The B cell compartment was profoundly depleted; however, IgD+ cells were nearly absent, and CD27- B lymphocytes were the majority of cells in the periphery. CD21 + B were also absent. The CD21- is a marker of some autoreactive B cells, which are expanded in a subset of patients with common variable hypogammaglobulinemia, as well as in autoimmune diseases and in the infant thymus [72,73]. The expansion of CD8 + CD28- accounts for the increase of CD8+ lymphocytes, a distinctive marker for Good’s syndrome, as indicated in the 2022 update of the inborn errors of immunity classification [74,75]. This subset of CD8 T cells is expanded in aged individuals, and it represents a marker of immune senescence [76,77]. In autoimmunity, CD4+ CD28- cells are expanded [78], but these were normally represented in our patient (see Table 1). There was a striking depletion of CD4+ T cells, and the patient tested negative for HIV on several occasions. This resulted in the inverted CD4/CD8 ratio, and an increase of CD3+ double negative T cells was noted. As for the thymic output as measured by coexpression of CD31, it was nearly undetectable for CD4+ and not measured in CD8+ cells; however, according to the CD45RA marker of naïve T lymphocytes, the majority of CD8+ cells were positive, but this may be explained by its reexpression in exhausted (CD28 negative) terminally differentiated T cells, and not because they are newly produced.

### 4.3. Autoantibodies to Cytokines

The inclusion of Good’s syndrome in the group of inborn errors of immunity named Phenocopies of PIDs put the presence of autoantibodies (and/or cell mediated responses to cellular components) as the main abnormality in this condition, driving the immune defects and the other associated features. Naturally occurring and disease-associated autoantibodies against several cytokines have been reported, including those to interleukin-1ß, IL-6, IL-8, IL-10, IL17, granulocyte-macrophage colony-stimulating factor, IFN-α, IFN-ß, IFN-γ, macrophage chemotactic protein-1 and IL-21 [79]. The exact role of these autoantibodies is not completely clear; however, it is thought that they participate in the autoregulation and homeostasis of the immune system as well as being pathogenic in some immunity defects, thus denominated autoantibody-mediated phenocopies of PIDs [16,74]. The cytokine network uses these natural autoantibodies to prevent excessive cascade effects for otherwise negligible stimuli [80] and thereby regulates T lymphocyte reactivity. A dysregulated network may trigger harmful autoinflammatory responses [81]. Such autoantibodies are enriched in immunoglobulin preparations for use in humans and may be partly responsible for their immunoregulatory properties [82]. However, some autoantibodies are induced in infectious diseases or, more probably, are increased in subjects who develop certain infectious diseases and are rendered more prone to them precisely because of neutralization of an essential component of their immune defenses, as in the case of chronic mucocutaneous candidiasis and anti-IL17 autoantibodies [67]. It has been reported that autoantibodies against IFN-γ, which trigger mycobacterial diseases and mimic inborn errors of IFN-γ immunity, are genetically determined [83]. The recent report that autoantibodies to IFN type I occur in severe cases of COVID-19 and account, in even a larger number of instances than for genetic defects of IFN-α production, for higher mortality especially in younger patients [84,85,86] has revived the interest in this neglected field of study. Autoantibodies to IFN-α in systemic lupus seem to contrast the pathogenic increase of type I IFN and characterize the course of the disease, with flares and remissions [87]. Autoantibodies to different chemokines have been found in HIV infection, autoimmune diseases, and also in patients suffering from post-acute sequelae of COVID-19 (long-COVID-19, [88]). Our finding of autoantibodies against three important cytokines in one case of Good’s syndrome should therefore pave the way to broader studies of anti-cytokine autoantibodies in a larger number of patients, in order to corroborate the allocation of this disease within the group of phenocopies of PIDs [89,90].

## 5. Conclusions

Despite the increase of reported cases of Good’s syndrome, its overall picture remains puzzling, due to the extreme clinical and immunological heterogeneity. The rarity of patients prevents in-depth examination of tissues (first of all, the thymus, but also bone marrow and the gut) and the use of appropriate techniques, sometimes unavailable or too expensive. Our case has been evaluated for both phenotypic lymphocyte markers and autoantibodies to six cytokines. The results may help to delineate multiple distinctive immune abnormalities shaping the clinical manifestations of the syndrome and contributing to the pathogenesis of this disorder.

## Figures and Tables

**Table 1 biomedicines-11-01605-t001:** Phenotypic characterization of absolute lymphocytes numbers/μL from the patient with Good’s syndrome.

Marker	Total Number/μL	Marker	Total Number/μL
CD3+	1003	CD3−CD16 + CD56+	52
CD4+	296	CD3 + CD8+	636
CD4 + CD45RA+	71	CD3 + CD8 + CD45RA+	444
CD4 + CD28+	223	CD3 + CD8 + CD28+	46
CD4 + CD25+	25	CD19+	15
CD4 + CD31+	1	CD24 + CD19+	0
CD3 + DN	73	CD21+	0
		CD19 + CD38+	10
		CD19 + CD27 + IgD+	0
		CD19 + CD27 + IgD−	2

Legend to Table 1. Absolute numbers of cells gated for the indicated marker or their combination. Explanation of type of lymphocyte subpopulation can be found in the text. Total numbers were obtained by the dual platform method (lymphocytes gated on lysed whole blood samples, white blood cell counts by hemocytometer). DN = double negative (CD4−CD8−).

**Table 2 biomedicines-11-01605-t002:** Clinical features of patients with Good’s syndrome (data in largest reported case series) for diagnosis.

Main Clinical Findings of Good’s Syndrome (in Order of Incidence)
Thymoma (mandatory)
Hypogammaglobulinemia (mandatory)
Sinopulmonary bacterial infections
Pure red cell aplasia
CMV infection (different organs)
Diarrhea, colitis
Candidiasis (mucocutaneous and other sites)
Lymphopenia, leukopenia, myelodysplastic syndromes
Urinary tract infections
Lichen planus

**Table 3 biomedicines-11-01605-t003:** Clinical manifestations presented by our patient (in order of time presentation). The absence of recurrent sinopulmonary infections could be due to early diagnosis of hypogammaglobulinemia and start of immunoglobulin replacement treatment.

Clinical Findings in Our Patient (in Order of Presentation or Diagnosis)
Hemolytic anemia
Persistent diarrhea
Hypogammaglobulinemia
Esophageal candidiasis
CMV (colitis)
Lichen planus (oral)
Thymoma
Oral candidiasis
Lymphopenia, aplastic anemia
Sepsis

**Table 4 biomedicines-11-01605-t004:** Types of thymoma in Good’s Syndrome, according to largest reported series.

Thymoma Type (Who Classification)	U.K. Series 2018	China Series 2017	Kelesidis Review 2010	*p* Significance
Type A	4	10 *	2	<0.01
Type AB	27	14	10	
Type B	6	4	10 *	<0.05
Type C	9 *	1	2	<0.01
Total Typed	46	29	24	

Table 4 shows the number of reported types of thymomas, sccording to the WHO classification; significant differences are indicated by *.

## Data Availability

Further information is available from the corresponding Author.

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
