# Peer review of "Insights from a Case of Good’s Syndrome (Immunodeficiency with Thymoma)"

_biomedicines, 2023, doi:10.3390/biomedicines11061605_

Round 1

Reviewer 1 Report

This is a well-written and detailed case report and also an excellent review of Good's syndrome. The readers will learn a lot and gain insight into this enigmatic disorder.

Minor: 

Page 3, line 114: comma after "infiltrate" should be period.

Page 5, Table 2: The reviewer thinks that "Main clinical findings ~" and "Clinical findings in" can be separated because the two findings are arranged in different orders. In addition, left alignment might be better for reading.

Page 6, line 244: Perhaps, not "lichen" but "lichen planus" is correct.

Page 7, lines 308-310: The reviewer would like to know the exact number of patients with each thymoma subtype, if available.

Author Response

We thank the reviewer for her suggestions for improvement. All corrections indicated have been made in the revised version. Table 2 has been divided into two distinct tables (new Table 2 and Table 3), in order to facilitate the readers, as recommended. The new Table 4 shows the numbers of thymomas in the three largest series reported, with regard to the type of thymoma (subtypes B1-B3 have been grouped together for easier comparison).

Reviewer 2 Report

I have enjoyed reading this very interesting case report on a rare syndrome. I have only minor remarks considering data presentation. In table 1 please add percentages in brackets behind absolute numbers. In table 2 signs and symptoms for syndrome and patient should be either presented in same order or somehow marked that patient fulfilled these symptoms/criteria. Otherwise paper is well written and interesting to read.

Author Response

We thank the reviewer for the positive comments. Table 2 has been changed into two separate Tables to facilitate the readers. The major criteria (both clinical and immunological-which is not listed in Table 2 which refers only to clinical manifestations) were fulfilled in our case, the other part of table (now Table 3) reported the temporal order of presentation of disease signs and symptoms, which varies greatly in different patients

In table 1 please add percentages in brackets behind absolute numbers.

We understand the rationale for giving both percentages and absolute numbers of subsets of lymphocytes when gated by the same field parameters (e.g. first gated on lymphoid cells by scatters, then tagged with monoclonal antibodies to CD surface antigens). However we then had a gating strategy different for each subset (e.g. focusing on B cells only, or on gated CD4+ T cells) so that the percentages would apply to a different total (where B cells or CD4+T cells now represent 100%) so that percentages would be confusing (being percentages of another percentage), and therefore absolute (CBC)- derived numbers are more clear and informative. This is the reason for giving uniform results as absolute numbers, since percentages refer to different gates and carry no further information.